# Adapting Railway Maintenance to Climate Change

**A. H. S. Garmabaki \*, Adithya Thaduri**  **, Stephen Famurewa and Uday Kumar**

Operation and Maintenance Engineering, Luleå University of Technology, 97187 Luleå, Sweden;
Adithya.thaduri@ltu.se (A.T.); stephen.famurewa@ltu.se (S.F.); uday.kumar@ltu.se (U.K.)
\* Correspondence: amir.garmabaki@ltu.se; Tel.: +46-70-614-69-16

**Abstract:** Railway infrastructure is vulnerable to extreme weather events such as elevated temperature, flooding, storms, intense winds, sea level rise, poor visibility, etc. These events have extreme consequences for the dependability of railway infrastructure and the acceptable level of services by infrastructure managers and other stakeholders. It is quite complex and difficult to quantify the consequences of climate change on railway infrastructure because of the inherent nature of the railway itself. Hence, the main aim of this work is to qualitatively identify and assess the impact of climate change on railway infrastructure with associated risks and consequences. A qualitative research methodology is employed in the study using a questionnaire as a tool for information gathering from experts from several municipalities in Sweden, Swedish transport infrastructure managers, maintenance organizations, and train operators. The outcome of this questionnaire revealed that there was a lower level of awareness about the impact of climate change on the various facets of railway infrastructure. Furthermore, the work identifies the challenges and barriers for climate adaptation of railway infrastructure and suggests recommended actions to improve the resilience towards climate change. It also provides recommendations, including adaptation options to ensure an effective and efficient railway transport service.

**Keywords:** climate change; climate adaptation; railway infrastructure; resilience of transport

## 1. Introduction

Railway infrastructure is the backbone for the growth and development of governments, businesses, and societies. Most of the existing railway infrastructure was designed and constructed several decades ago to fulfill society's mobility demands, including freight transport. For instance, in Sweden, most of the infrastructure was built in the early 19th century. Due to an exponential increase in population and demand for transport network infrastructure, the current infrastructure is experiencing loads higher than that of the designed capacity limits [1]. Transport infrastructure is also exposed to extreme weather events and climate change evolution, creating more excessive deterioration [2]. Considering practical constraints related to capital investment, government policies, and sustainability issues of building new railway infrastructure [3], utilizing climate adaptation options on the operation and maintenance of existing railway infrastructure is inevitable [4,5]. Adaptation options are measures and actions that can be implemented to improve adaptation to climate change. Climate adaptation for railway networks refers to the process by which traffic administration, including infrastructure and rolling stock, should mitigate and control risks due to extreme weather events and gradual degradation of infrastructure [6,7].

In the past decades, it has been observed that railway infrastructure is vulnerable to extreme weather events related to temperature, rains, floods, rising sea levels, frozen soil, etc. [8–10]. Extreme winds and storms can damage coastal railways, cause snowdrift during winter on railway infrastructure, and interrupt rail operations. Heatwaves can lead to rail buckling and damage to railway infrastructure, thereby causing traffic interruption or catastrophic derailment failure in extreme cases. The hot and dry conditions also may lead to vegetation and forest fire due to ignition caused by rail and rolling stock

wheel contacts. Furthermore, permafrost distribution and the freezing and thawing cycles can cause frost heaves at railway infrastructure in the Arctic region, damaging bridges and building foundations and their load-carrying capacity [11]. These events reduce the dependability performance of the infrastructure (reliability, availability, maintainability, and safety (RAMS)) [12–15].

Several past studies and research projects are making progress to assess the impact of climate change on railway infrastructure [16–21]. These studies recommend updating the national policies and regulations and integrating climate change and climate adaptation policies throughout railway infrastructure's various life cycle stages. Accordingly, there is also a need to revise the required maintenance and renewal policies to deal with climate changes and climate adaptation for short- and long-term scenarios to ensure robust and resilient infrastructure [20,21].

However, the findings of these investigations are not being applied by the railway stakeholders because of numerous challenges and barriers, such as knowledge and awareness about climate change impacts, shortage of capital investment, the nonexistence of policies, shortage of resources and time, etc. [22,23]. Moreover, it is difficult to comprehend the impacts of climate change since it changes with geographical locations, and in many cases, there is a lack of studies of risk consequences and impact assessments for the region. Therefore, there is a need to design and develop an effective and efficient roadmap to control climate change impacts. This can be achieved through a holistic perspective considering local climate conditions, climate change impacts, stakeholders' participation, policy changes, and infrastructure health during the operation and maintenance phase [24].

The main climate parameter that is responsible for climate change is temperature. It is reported that global temperature will rise by 1.2 °C and the current mitigation actions are not satisfactory for all nations to achieve net-zero emissions by 2050–2060 [25]. The feasible approach is to utilize the climate adaptation strategies to control climate change impacts. In the EU (European Union) region, disaster risk reduction utilizing climate adaptation options has been received as one of the top priority research areas in the transport network and urban built environments [3,26,27].

Climate adaptation is a complex and complicated approach that necessitates the understanding of various interdependencies between climate parameters and infrastructure assets using models for the prediction of the health of the asset [28]. Thaduri, Garmabaki [29] also provided a state-of-the-art review of these quantitative models on railway infrastructure. These studies identified the various vulnerable assets within railway infrastructure, such as tracks [2,30–33], turnouts [34,35], embankments [36–38], bridges [39,40], drainage systems [41], etc. These vulnerable assets are confined to their geographical location, and its potential consequence of failure depends on the asset's criticality, usage profile, and design limitations. Due to the lack of quantitative data representing the above interdependencies, several researchers and projects have studied the impacts of climate change on railway infrastructure by applying various qualitative approaches [9,42–44].

Climate risk assessment is a prerequisite for planning a roadmap to mitigate climatic impacts and identify critical infrastructure assets and vulnerable geographical locations. Several studies highlighted that temperature [2,45–47] rainfall [39,48,49], humidity [49,50], snow [30,38], permafrost [38,51], storms [52], and sea-level rise [53] are the major climate parameters that could impact the railway infrastructure [29]. However, most of these climate parameters are indeed specific to geographical location and severity, and the occurrence of these events and changes may vary among countries and/or regions. The impact of these climate change parameters on vulnerable railway assets leads to several consequences such as track movement, track buckling, track washout, erosion of trackbed, overflooding, falling of trees, higher winds, visibility, drainage system clogging, landslips, disruption of bridge foundations, settlement of edifices, arcing of conductive components, wayside fires, vegetation, etc. [29,49,54,55]. Since climate change is also a major concern for governments, it is imperative to plan and implement policies and regulations so that the stakeholders can accordingly build their strategy to handle climate change impacts [56,57].

Finally, several methodologies have been implemented for climate adaptation of railway infrastructure in extreme weather events and climate change variation [22,43,58]. Hence, we also need to discuss and develop methods and methodologies to implement climate adaptation procedures for various stakeholders within the railway transport sector.

Conducting qualitative analysis is a rigorous and time-consuming process; for instance, see Canada [59], UK [23], USA [60–63], Germany [64], Tanzania [65], Malaysia [55], Baltica [66], and Sweden [56] that are not entirely focused on research gaps identified above from the operation and maintenance of the railway.

Our previous work has carried out an overview of climate change impacts from Canada, the U.K., USA, and Asia [29]. From these studies, it was found out that there is a varying degree of impact of climate change on transport infrastructure because of geographical location, $CO_2$ emissions, and condition of the infrastructure. For example, countries such as the U.K., Sweden, and some parts of China are prone to extreme weather events from snow, low temperatures, and permafrost. Accordingly, there is also a need to implement climate adaptation measures for railway infrastructure. To understand the correlation between all the extreme weather events and their effects on railway infrastructure, it is necessary to conduct quantitative analysis. An alternative way is to conduct qualitative analysis, which can also serve as an input for conducting quantitative analysis [67].

Hence, this paper aims to investigate the impact of climate change and extreme climate on the operation and maintenance activities of the Swedish railway infrastructure. This study was carried out by conducting a comparative questionnaire study and interviews with relevant railway stakeholders.

This paper is structured as follows. Section 2 deals with adaptation and maintenance development debt of transport infrastructure and its urgent action. The research methodology and design of the questionnaire are discussed in Section 3. Section 4 represents the survey analysis and impact assessments of climatic change on the railway in the past 10 years. Furthermore, questionnaire descriptions and related features are discussed in this chapter. Finally, Section 5 provides the conclusions and remarks.

## 2. Adaptation and Maintenance Development Debt of Transport Infrastructure

Increased intensity and frequency of extreme weather events caused by climate change have various adverse impacts on rail infrastructure service performance and related costs. Research has shown that adverse climate conditions are responsible for 5 to 10% of total failures and 60% of delays on the railway infrastructure in northern Europe. The transport infrastructure managers have been striving to keep up with the increasing maintenance rate required for transport networks following corrective actions. For instance, the Swedish government expects that the appropriation of operation and maintenance of existing railway infrastructure will increase by 47 percent in the forthcoming years [68]. The excessive cost and failure consequences necessitate infrastructure managers to consider climate change actions in their business plan; however, the speed of such activities is not promising.

The gap between the adaptation performed and the required adaptation for transport network is called "adaptation and maintenance development debt". An explanatory model of the discrepancy between the required adaptation and maintenance activities and the actual status of transport adaptation and maintenance, in terms of resources, tools, and techniques is visualized in Figure 1, [69]. The gap between the status of "adaptation and maintenance" and the "adaptation and maintenance required" for transport infrastructure is simulated using a typical linear regression model.

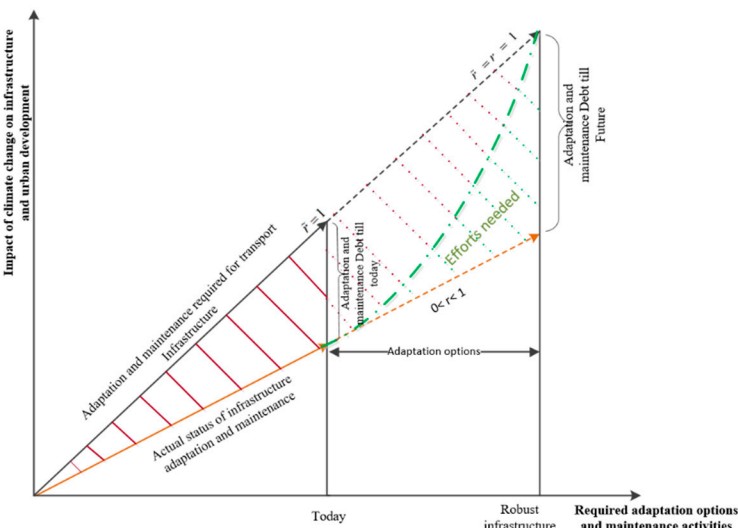

**Figure 1.** Adaptation and maintenance development debt.

Pearson's r is nominally used to represent the correlation between two variables, namely, the impact of climate change on transport infrastructure and urban development and the required adaptation and maintenance. The association between the impact of climate change and adaptation required for transport infrastructure (black line) is assumed to be a total positive linear correlation ($\tilde{r} = 1$). The association between the impact of climate change and the actual status of infrastructure is also positive, but with a lower rate: $0 < r < 1$, [69,70].

The discrepancy between the required adaptation and maintenance and the actual status of adaption and maintenance of transport infrastructure created the current development debt (the red dashed area in Figure 1), which has increased with accumulated interest. Efforts are needed to close the adaptation and maintenance debt (the green dashed area in Figure 1) and reach a robust infrastructure where transport infrastructure's required adaptation and maintenance are reconnected with the actual status of adaptation and maintenance, i.e., where $\tilde{r} = r = 1$. If the current adaptation policies continue in the same manner, the debt will continue to increase, leading to more climatic traffic disruption, loss of resources, unsafe transport, etc.

### 3. Research Methodology

The research methodology is presented in Figure 2. This study employs a questionnaire survey to gather appropriate information for describing the impact of climate change on railway transport infrastructure. This study seeks to assess the perception of climate change impacts on Swedish railway infrastructure and to better understand the causal relationships between climate parameters and infrastructure conditions. These assessments can help evaluate the impact of climate change on railway maintenance and propose relevant adaptation policies. The objectives of this questionnaire from the identified research gaps are:

- To assess the level of awareness of climate impacts on the railway sector within society;
- To identify the climate parameters that could impact the state and maintenance need of railway infrastructure; and
- To identify diverse types of risk incidences connected to climate change and consequences.

The first four activities illustrated in Figure 2 are presented in the previous sections and the other remaining activities are described in the subsequent sections below.

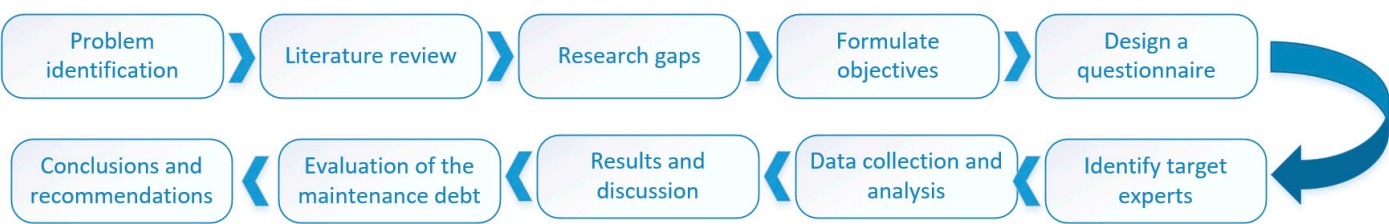

**Figure 2.** Illustration of the proposed research methodology.

### 3.1. Design of the Questionnaire and Identify Experts

For the present study, a questionnaire survey was designed, adopted from a UN questionnaire study [71] and consisting of 38 questions (see Supplementary Materials). This questionnaire was distributed to several experts at the Swedish Transport Administration (TRV) and the Swedish Transport Agency (Transportstyrelsen), Swedish municipalities, train operators, and other maintenance service providers, and 55 responses were received. The general description of the questionnaire and its structure is summarized in Table 1. The detailed description and analysis of these responses are presented in subsequent sections.

**Table 1.** Description of the questionnaire and its structure.

| No. | Questions | Description |
|:---:|:---:|:---:|
| 1 | Q1–Q7 | General description of the experts and their domain of expertise |
| 2 | Q8–Q11 | Awareness of climate change on railway transportation infrastructure |
| 3 | Q12–Q17 | Impacts of extreme events caused by climatic change on railway in the past 10 years |
| 4 | Q18–Q20 | Data collection and information strategy |
| 5 | Q21–Q28 | Identification and assessment of several types of risks, consequences (cost) |
| 6 | Q29–Q38 | Measures, activities, and policies for climate adaptation with references to maintenance |

### 3.2. Identification of Experts

In this questionnaire study, several experts were asked to share their opinions and experiences concerning climate change impacts on the transport system, its associated failure modes, and failure consequences. The processes for expert selection and expert opinion elicitation are discussed in the following steps.

#### 3.2.1. Step I: Expert Selection

This study utilizes expert judgment approaches to identify different impacts of climate change and performed a qualitative analysis of the consequences of climate-based failures on railway infrastructure. This assessment can be considered subjective due to its dependence on experts' opinions, knowledge, experience, and selection process.

The selection of experts is a major step in the study as it can affect the veracity and reliability of the research outcome. The term "expert" used in this study is according to the definition provided by Naseri and Barabady [72], Otway and Winterfeldt [73]: "a person who has a background in the subject matter at the desired level of detail and who is recognized by his/her peers or those conducting the study as being qualified to solve the questions". The experts who participated in this questionnaire are Swedish transport infrastructure managers, maintenance organizations, train operating companies, and municipalities (road and railway). Table 2 shows the job title of the respondents. The ratio of the gender of the respondent is 40 percent for women versus 60 percent for men.

**Table 2.** Area of expertise of participants.

| Project Leader | Energy, Environmental, Sustainability Strategist | Climate Adaptation Expert | Community Planning, Planning Strategist | Voice Director, Manager | Infrastructure Engineer, Railway Specialist | Operations Manager, Technical Manager |
|---|---|---|---|---|---|---|
| 2 | 12 | 5 | 5 | 7 | 20 | 4 |

Figure 3 depicts the active railway lines in Sweden, which is an important aspect in the selection of an expert. Figure 4 represents the geographical distribution of the respondents. In some regions, there was more than one respondent for the questionnaire. As given in Figure 4, the study was not restricted to any specific geographical location, and the respondents were distributed across the whole country. It is noteworthy that about 20% of experts have national responsibilities, and the rest are responsible for their region and municipalities.

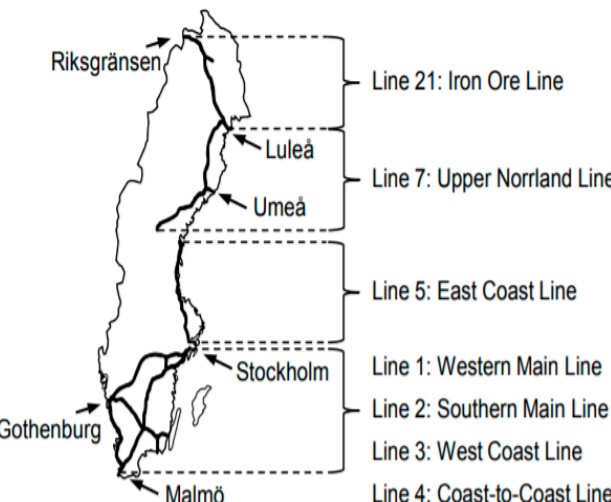

**Figure 3.** Swedish railway network.

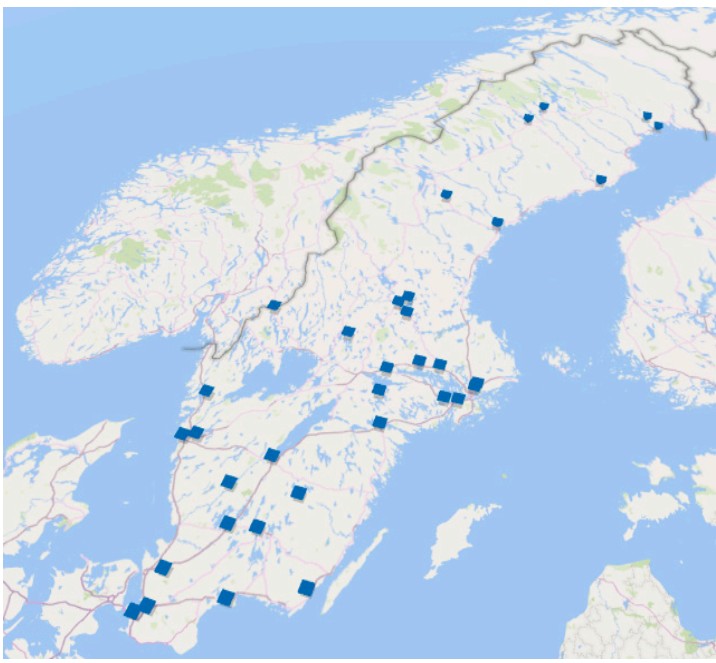

**Figure 4.** Geographical distribution of the respondents.

3.2.2. Step II: Expert Opinion Elicitation

This process was the second step in the expert judgment analyses and is defined as "the process of obtaining the subjective opinions of experts through specifically designed methods of communication, such as surveys, interviews, group meetings and questionnaires" [74]. A qualitative approach in the form of a questionnaire was used here for expert opinion elicitation. In qualitative approaches, the experts are asked to express their opinions about a parameter in the form of, for instance, an absolute rating, distribution approximation, interval scaling, etc. [75]. In addition, to check the reliability of the questions, Cronbach's alpha was calculated. This index was calculated for some questions by SPSS (Statistical Package for Social Sciences) software and the outcome is located between an acceptable and good consistency range, i.e., between 0.7 and 0.9.

## 4. Results and Discussion

The results and discussion are presented under five main categories in line with the objectives of the studies. Furthermore, to assess the responses, the outcome from the UN questionnaire [71] was considered as a comparison for questions with similar objectives.

### 4.1. Awareness of Climate Change and Its Impact on Railway Transportation Infrastructure

Based on our literature study, awareness has been identified as the key factor in the climate adaptation process. Figure 5 presents and compares how climate change can act as a problem in railway transport networks in the UN in 2013 and Sweden in 2020. Of the questionnaire respondents in Sweden, 40% believe that climate change can have a "High" impact on transport networks, while their perception of the "Very high" impact of climate is about 10%, which is half of that of the UN respondents [71]. The distribution respondents' rank is left-skewed based on the statistics, and this means that only a few experts have the confidence of less impact of climate change (Q8).

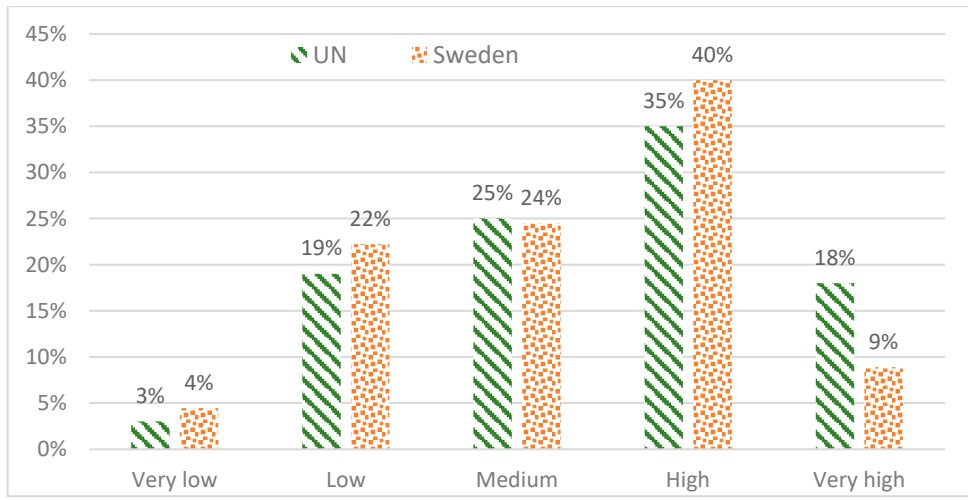

**Figure 5.** Climate change identified as a problem in railway transport networks among Swedish and UN survey respondents.

The statistics show that the respondents tend to see climate change impacts in a brief period, whereas almost 80 percent of the UN survey respondents mentioned that climate change problems would persist over the next 30 years (Q9). Moreover, about 65% of the respondents considered that climate change's impacts would be revealed in a time scale between 0 and 20 years (near future time scale), and 20 percent considered that the impacts would be a problem in less than 10 years. On the other hand, about 20% believed that the impacts could lead to problems in the next 20 years (see Figure 6).

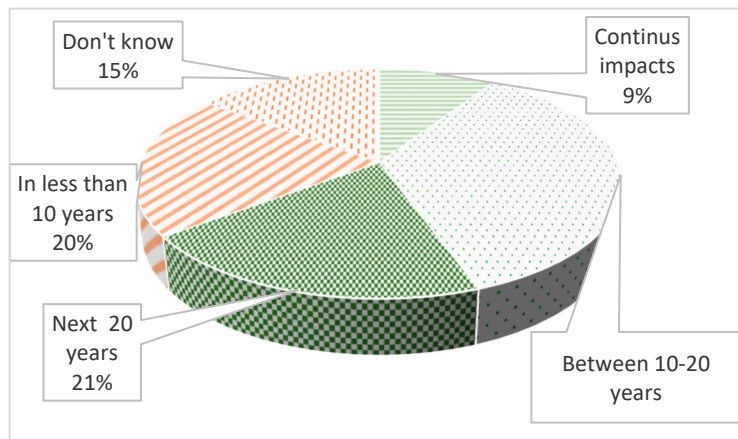

**Figure 6.** Time scale for the presence of transport problems due to climate change.

Dedicated educational programs and training are essential aspects of global response required to raise awareness of climate change. It helps society recognize the effect of global warming and helps them deal with climate change impacts and implement adaptation policies [76]. To increase the level of awareness, it is important to identify the main target audience. The answers showed that the main target audience was government and local authorities, including the Swedish Transport Administration (TRV) and Swedish Transport Agency (Transportstyrelsen), state and regional politicians, etc., as given in Figure 7 as per respondents from Table 1. Excluding the categories mentioned above, other audiences such as manufacturers, public society, and contractors are recognized at a low level (15%) as the main target audience for raising awareness of climate change impacts. This aligns well with the UN study, as respondents also mentioned that the main responsible organizations are public and local authorities, ministries, and other government bodies (Q10).

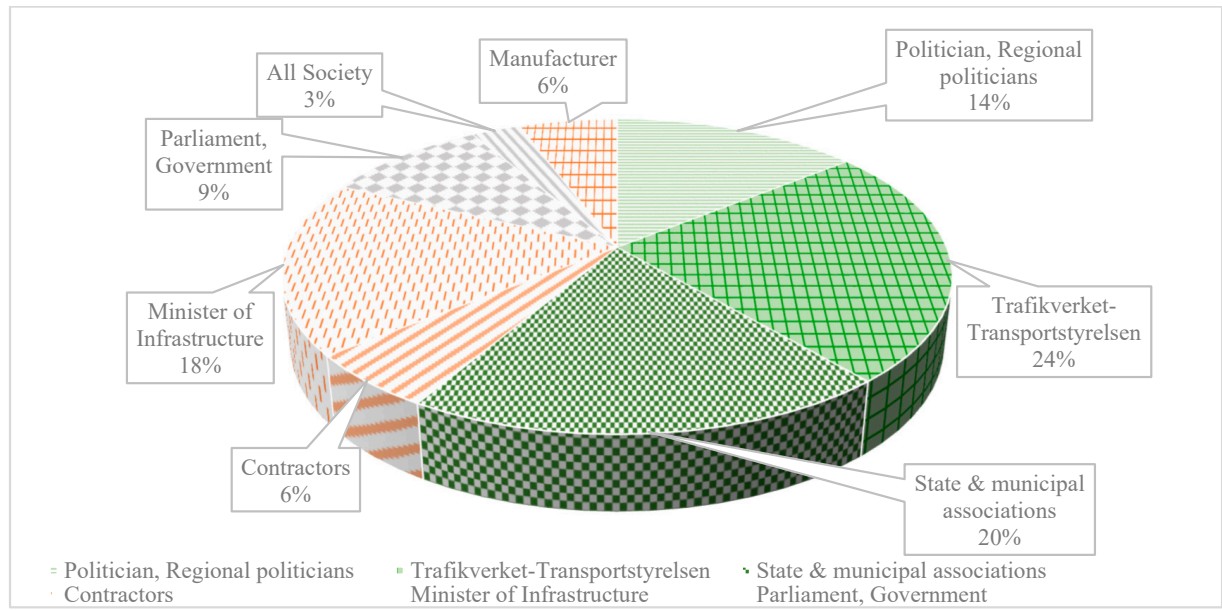

**Figure 7.** The main target audience for raising awareness about climate change impacts on railway infrastructure.

A question was asked to evaluate the level of awareness of society of any extreme weather or climate impacts (rising temperatures, extreme events, droughts, floods, sea level rise, storm surges, melting permafrost, etc.) on railway infrastructure in the past 10 years. As given in Figure 8, about 50 percent of respondents were aware of a medium to the very high impact of climate change. It is noteworthy that 23 percent of respondents selected

the "Don't know" alternative, which confirms that there is a need to increase society's awareness about climate change impacts on the rail transport system (Q11).

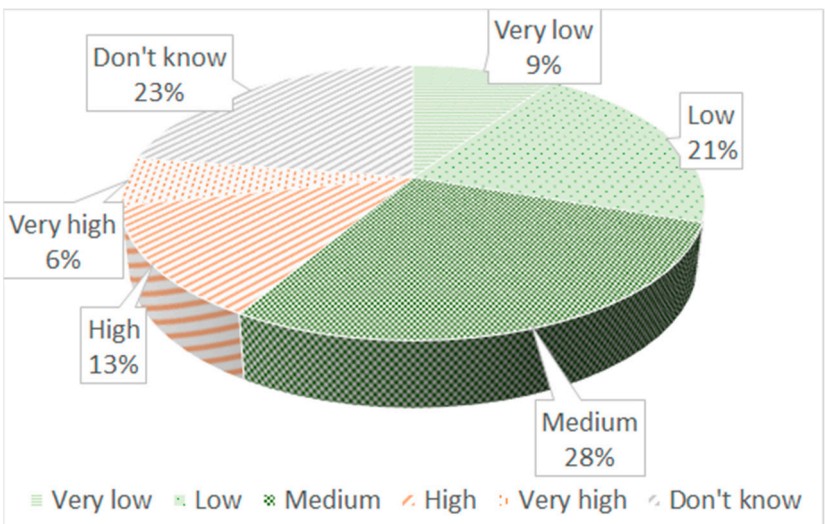

**Figure 8.** Extreme weather or climate factors (rising temperatures, extreme events, droughts, floods, sea level rise, and storm surges, melting permafrost, etc.) impacts in the past 10 years.

Furthermore, respondents were asked about previously conducted studies on the impact of climate change on the railway network at their regions/organizations (see Figure 9). One-third of the respondents mentioned that they had performed such studies, and about one-third replied either "Don't know" or "Blank", emphasizing that there is a need to increase the level of awareness. The outcome also reveals that 35% of the regions/organizations did not conduct any study, which may be related to lack of awareness, trust, or other barriers (Q12).

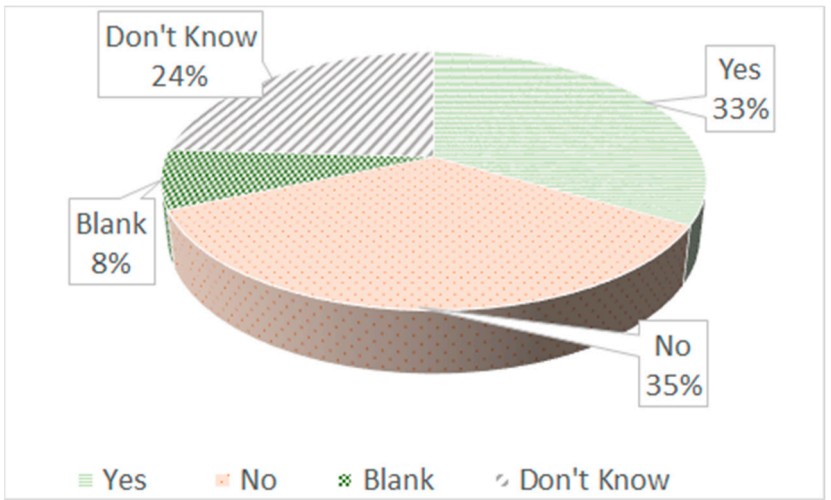

**Figure 9.** Conducted studies evaluating the implications of climate impacts on railway/transport infrastructure.

### 4.2. Impacts of Extreme Weather Events on Railway Services in the Past 10 Years

Comprehensive questions have been designed to identify the severity of the impact of climate change on railway operation and maintenance within the questionnaire. Extreme weather events such as temperature, flooding, extreme wind, lightning, frozen soil, and permafrost have been selected, which are relevant for Sweden. Furthermore, to assess the impact of climatic changes on the operation and maintenance of the railway network,

the following five key indicators of railway operation and maintenance services have been considered.

- Railway infrastructure robustness;
- Railway operation;
- Railway safety;
- The economic impact for various stakeholders; and
- Other vulnerability/impacts.

The outcome of the survey is categorized in the following subsections.

### 4.2.1. Temperature Impact

As shown in Figure 10, most respondents emphasized that temperature has a high impact on the different key indicators of railway operation and maintenance services, except for the category identified as "Other vulnerability" due to lack of information and awareness.

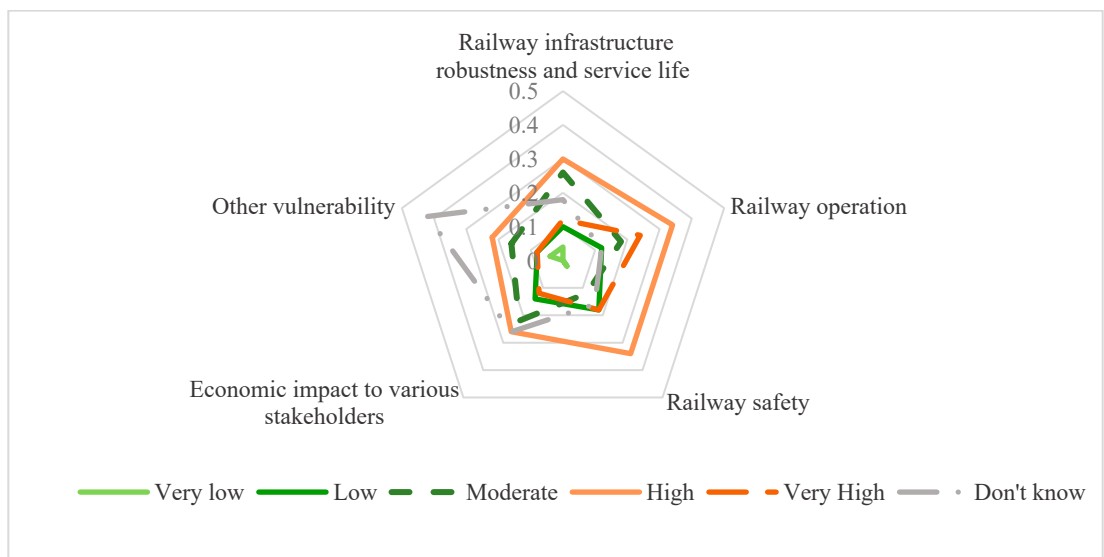

**Figure 10.** Impact of temperature on the different railway operation and maintenance services.

To summarize the statistics given in Figure 10, the medium, high, and very high impacts have been aggregated and interpreted as "significant impact". Thus, the significant impact of temperature on "Railway infrastructure robustness and service life" is about 70%. In the same way, the significant impacts of temperature on "Railway operation", "Railway safety", and "Economy impacts to various stakeholders" are 68%, 64%, and 60%, respectively (Q13).

### 4.2.2. Heavy Rains/Flood Damage/Drainage Impact

Similarly, respondents considered that flooding has a significant impact (aggregation of moderate, high, and very high levels) on "Railway operation". The significant impact of climate change on "Railway operation" is about 73%. The underlying reasoning for this high score is the fact that flooding can cause soil erosion underneath the rail track or submerge in the track superstructure due to an inadequate water drainage system. The additional impact of flooding of railway infrastructure is overhead contact lines that can cause short circuits of the signaling system. In some cases, if water levels rise above the rails, the train operator needs to reduce speed to prevent damage to the train and infrastructure, which in turn can cause delays.

The significant impacts of flooding on "Railway infrastructure robustness and service life", "Railway safety", and "Economy impacts to various stakeholders" are 67%, 68%, and 63%, respectively. The "Other vulnerabilities" alternative in the figure below received a

high rank due to lack of information/awareness and broad perspective and the unclear nature of the question; see Figure 11 (Q14).

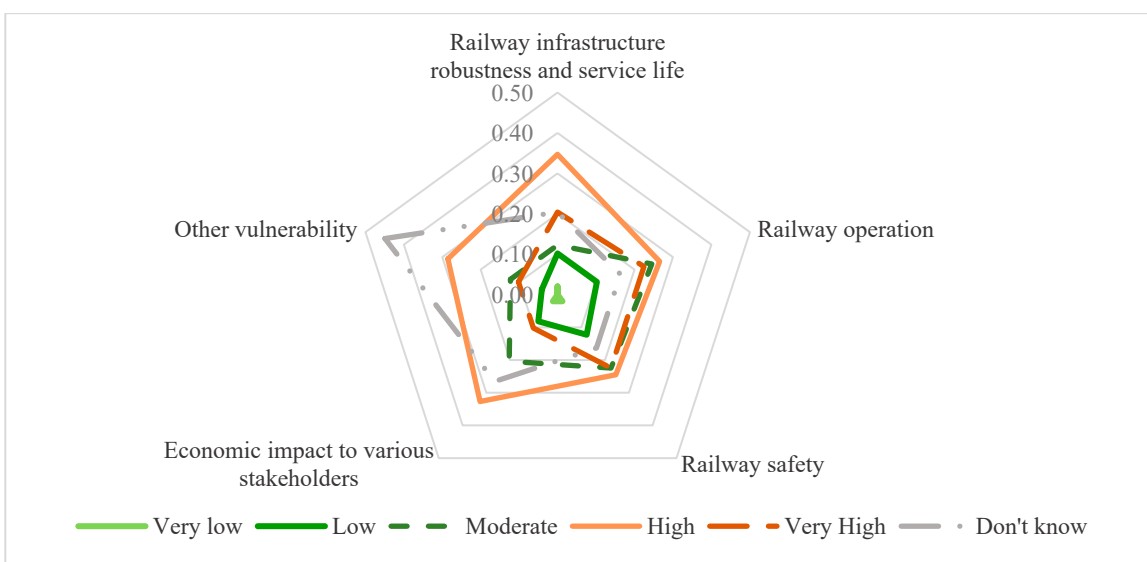

**Figure 11.** Impact of flooding on the different railway operations and maintenance services.

### 4.2.3. Storms/Extreme Wind Impact

Storms and extreme winds have a significant impact score of 50% on "Railway infrastructure robustness and service life". However, this impact is lower when compared to the rated impact of temperature and flooding. The significant impacts of storms and extreme winds on "Railway operation", "Railway safety", and "Economy impacts to various stakeholders" are 71%, 58%, and 52%, respectively. About one-fourth of experts chose the "Don't know" option related to railway operation and services, a confirmation of a lack of information, studies, and awareness among respondents; see Figure 12 (Q16).

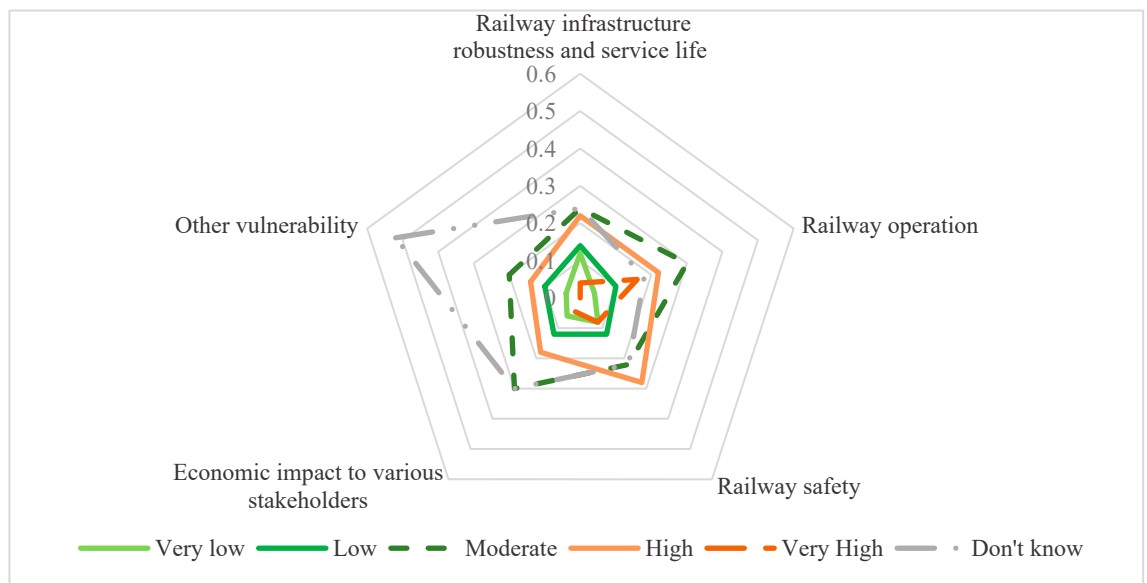

**Figure 12.** Impact of storms/extreme wind on the railway operation and maintenance services.

### 4.2.4. Lightning Impact

Figure 13 shows the impact of lighting on selected key indicators of railway services. Signal systems, including onboard and trackside devices, are especially vulnerable to

lightning and electromagnetic interference due to the sensitivities to radiation, electric and magnetic fields. Approximately 40% of the respondents selected "Don't know" for all alternatives. This value is due to the low level of awareness and information related to lighting's impact on railway services. The significant impact of lightning on "Railway operation" is about 44% and is considered as moderate. The significant impacts of lightning on "Railway infrastructure robustness and service life", "Railway safety", and "Economy impacts to various stakeholders" are 26%, 32%, and 36%, respectively (Q16).

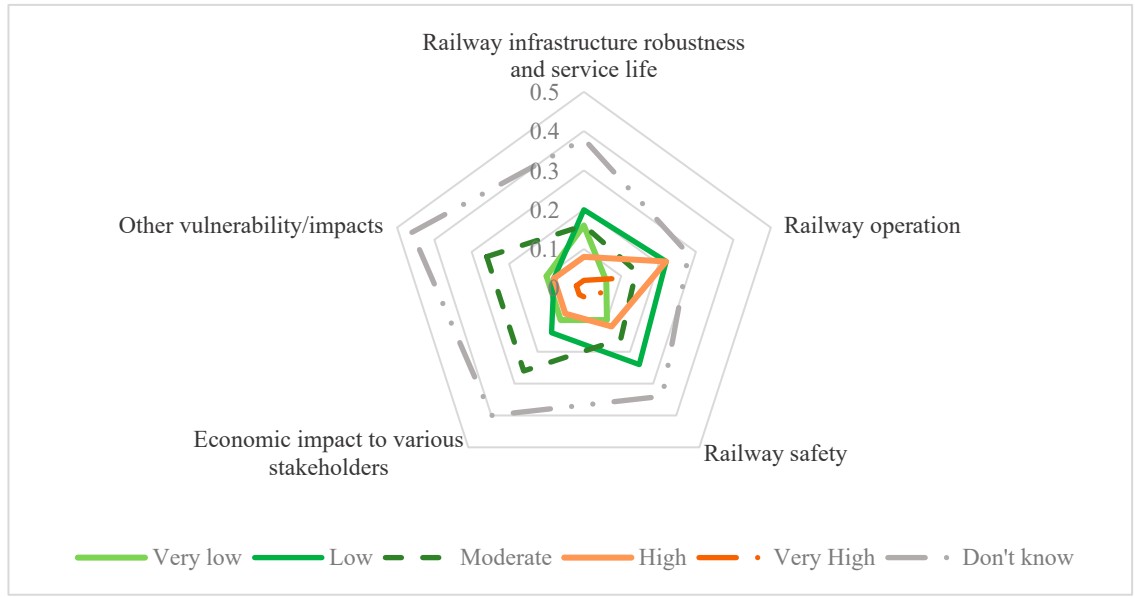

**Figure 13.** Impact of lightning on the different railway operations and maintenance services.

### 4.2.5. Frozen Soil Impacts

Frozen soil impacts are critical and can cause lateral and vertical movements on the railway system, especially in Nordic countries. Most of the experts believed that "Railway infrastructure robustness and services" and "Railway operation" will be affected by frozen soil at a "High" weightage of 35% and 33% (Non aggregated rate), respectively, given in Figure 14.

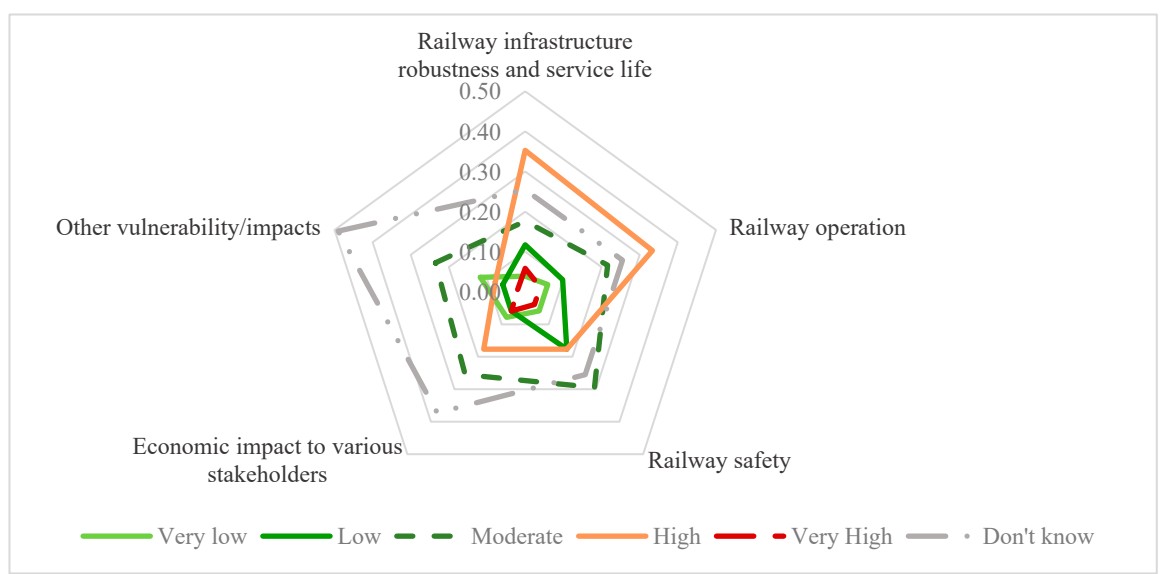

**Figure 14.** Impact of frozen soil on the different railway operation and maintenance services.

The significant impacts of frozen soil on "Railway infrastructure robustness", "Railway operation", "Railway safety", and "Economic impact" is 59%, 59%, 51%, and 49%, respectively. Most of the respondents rated frozen soil as "Moderate" or "High" as an impact on railway operation and maintenance services, due to the increase in air temperature and reduction in snow coverage on the ground during cold seasons in Nordic countries. In addition, several respondents selected the "Don't know" alternative, highlighting the need to perform further study or improve awareness of such impacts within society (Q17).

### 4.3. Data Sources, Data Availability, and Analytical Tools

To control, measure, and assess the impact of climate change on railway infrastructure health, there is a need to collect data related to the occurrence of extreme weather events and the long-term evolution of climate change and their impacts. As presented in Figure 15, the respondents identified the Swedish Metrological and Hydrological Institute (SMHI), research institutes, and universities as the main sources of information (38 percent) to study climate change's effects. Other public authorities and national agencies such as the Environmental Protection Agency and County Administrative boards of Sweden are identified as sources of relevant information for climate change and its impacts. Some of the respondents selected metrological information from national and international domains as a source of information. The statistic of the data source for the investigated stakeholders is presented in Figure 15; metrological data from other sources besides SMHI are classified as "Metrological info" (Q18).

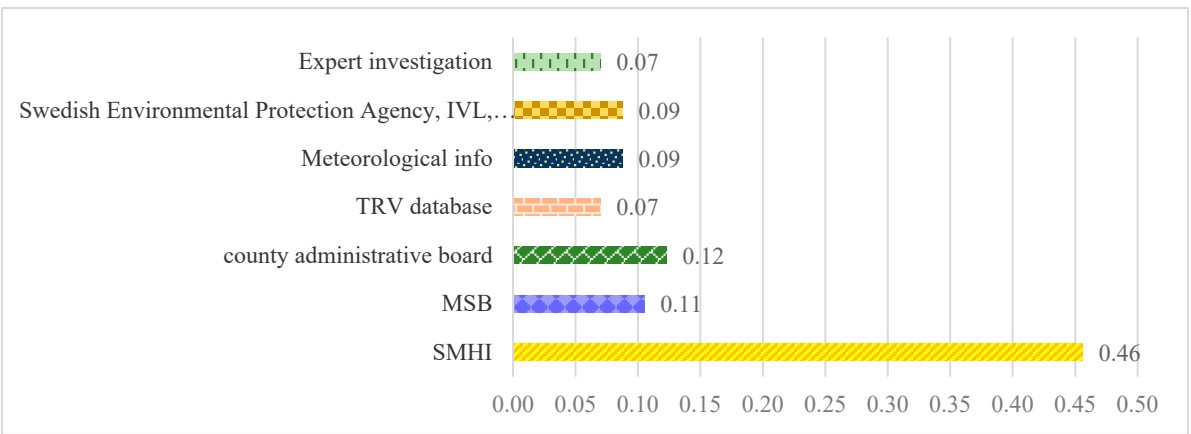

**Figure 15.** Main information sources used to study the effects of climate change.

It is noteworthy that all the above information sources mentioned by the respondents are related to the national agencies. Furthermore, companies' roles in climate adaptation, including manufacturers, constructors, contractors, and operators, are not addressed by experts. Moreover, TRV, the main infrastructure manager in Sweden, is not highly ranked as the main information source, although it is perceived as an important player dealing with climate adaptation and related responsibilities in the previous questions.

In addition, a question related to the availability of climate change and climate impact data for transport infrastructure was asked. The distribution of opinion of the respondents is almost concentrated at a "Medium" level, and a few experts rated it as either "Low" or "Very good" (about 7 percent). Only 2 percent of experts selected "Excellent" data availability. As illustrated in Figure 16, a similar trend can be seen compared to the outcome of the UN survey, except for the "Don't know" alternative for the Sweden case, which is due to a lack of information (Q19).

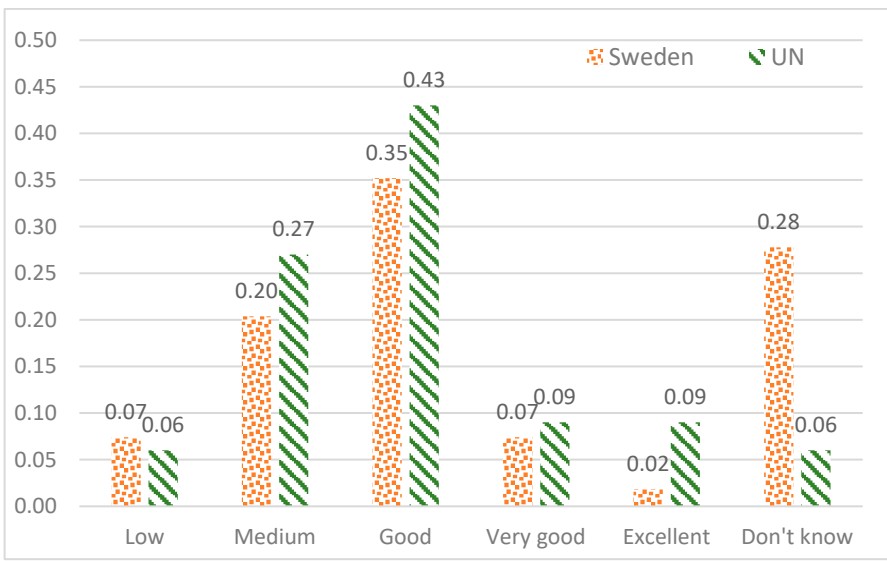

**Figure 16.** Availability of relevant data/information.

It is also interesting to know about the tools that the experts are using to detect, predict and assess critical weather-related infrastructure incidences. The result revealed that more than 60 percent of the respondents were unaware of operation models and software tools in use at their organization. This observation is not surprising, since interest in climate change and initiatives to mitigate its impacts in different fields such as research, finance, and infrastructure is new. However, this gap needs to be addressed urgently to increase the awareness and use of appropriate software tools. This action can be accelerated by further collaboration, cooperation, and broader dissemination of available information and best practices regarding relevant tools. Some experts utilize Global Information System (GIS) analyzers, and the tools provided by SMHI and the other tools mentioned by individual respondents are provided in Figure 17 (Q20).

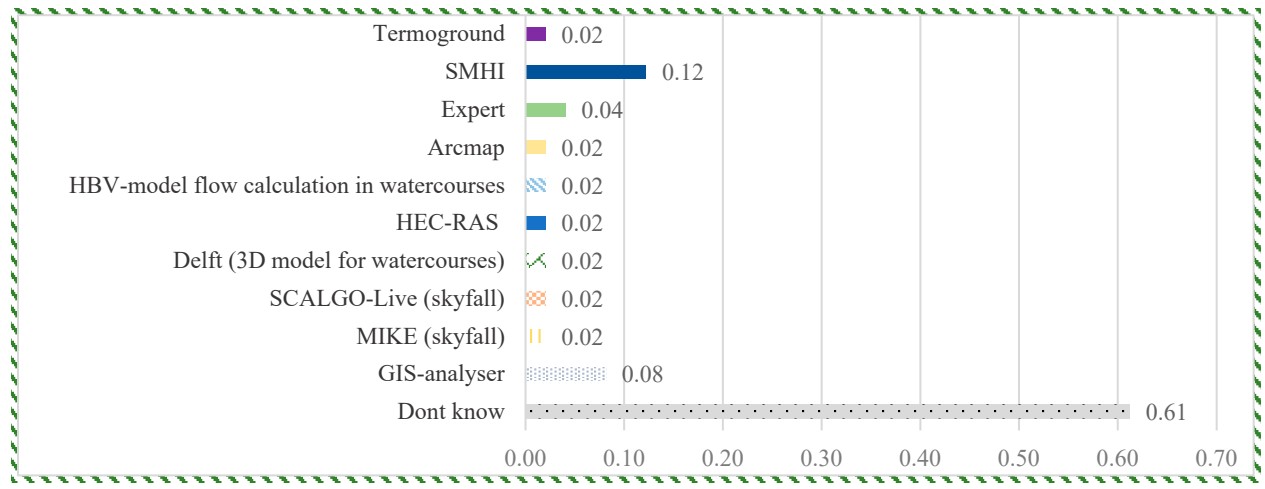

**Figure 17.** Operational models/software tools that region/organization used for the prediction of weather-related risks to critical infrastructure.

### 4.4. Identification and Assessment of Several Types of Risks and Consequences

Cost–benefit analysis is a crucial tool for climate change impact assessment and selecting suitable adaptation plans or strategies. About 40 percent of experts were unaware of the cost–benefit methodology for assessing potential damage due to climate change and selecting suitable adaptation plans in their region/organization. Moreover, 50 percent of respondents mentioned that such an analysis had not been conducted in their organiza-

tion/region, and only 9% have experience using cost–benefit or similar approaches for selecting an appropriate strategy for climate adaptation. As illustrated in Figure 18, in comparison to the UN study, respondents from Sweden have conducted far fewer studies. Hence, there is a need to perform such analyses at a regional level to reduce the adaptation and maintenance development debt (Q21). In addition, about 80% of those who implement the cost–benefit analyses scale applying cost–benefit analysis at an elevated level (Q22).

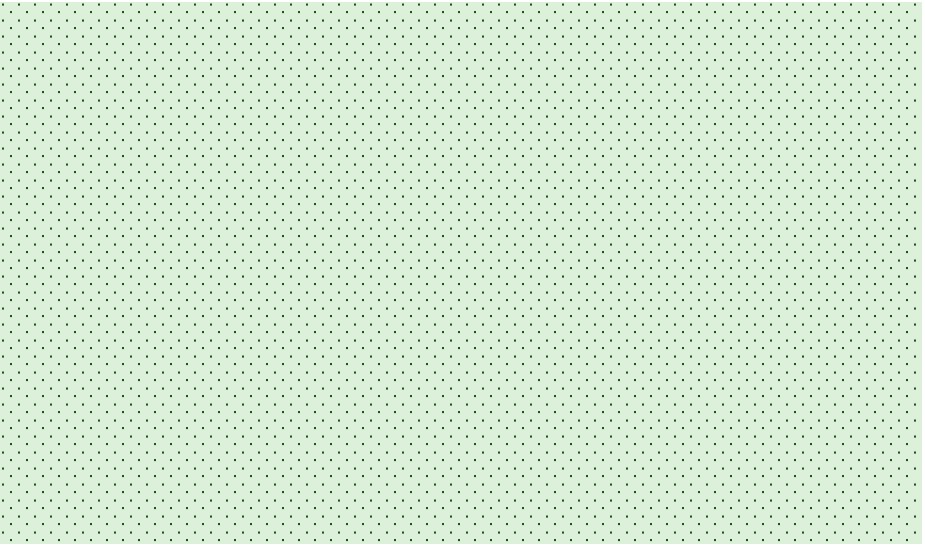

**Figure 18.** Comparison rate of conducted studies to estimate the costs of actual or potential damage due to climate change, e.g., utilizing cost–benefit analysis.

There is a need to map the level of awareness of different vulnerabilities, risks, and associated consequences within the railway transport network due to extreme events. The survey shows that the awareness level in Sweden regarding current railway/transport infrastructure vulnerabilities to natural hazards is quite low, about 20 percent. Furthermore, about 20 percent of the respondents have an opinion other than railway infrastructures are not vulnerable to climate change. Moreover, 22 percent are well informed of the vulnerabilities of transport infrastructure to climate change. Among those who acknowledged the vulnerability of climate change, 60 percent performed risk-based approaches to identify the vulnerabilities. Figure 19 compared the rate of awareness between UN and Sweden studies. This result confirms an urgent need to disseminate information and knowledge about potential vulnerabilities within public administration/organizations and stakeholders in Sweden (Q25, Q26, Q27, Q28).

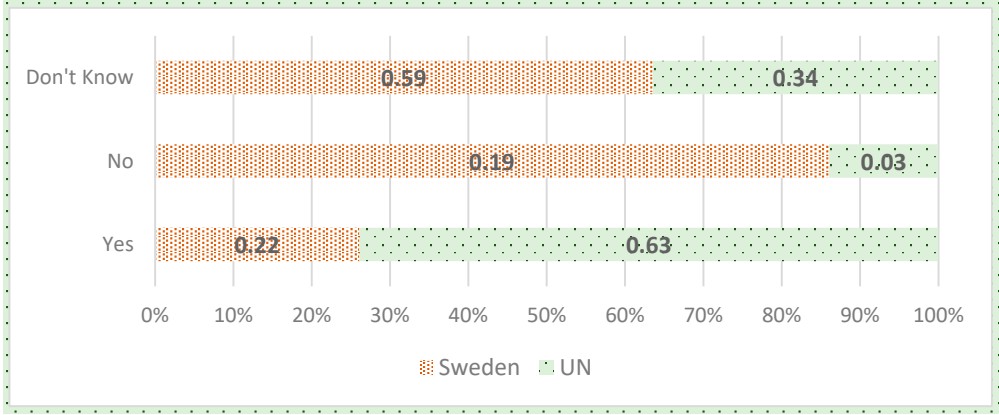

**Figure 19.** Level of awareness among public administrations/organizations in Sweden and UN survey respondents regarding current vulnerabilities of railway/transport infrastructure to natural hazards.

The results in Figure 20 were extracted from the experts' opinions regarding their practice of vulnerability assessment of railway/transport infrastructure due to different extreme climatic factors. As reported in previous questions, the rate of stating the "No" alternative was higher than other alternatives except for flooding. The analyses showed that on average, about 25 percent of respondents implemented specific assessments for weather and/or climate change factors (Q23).

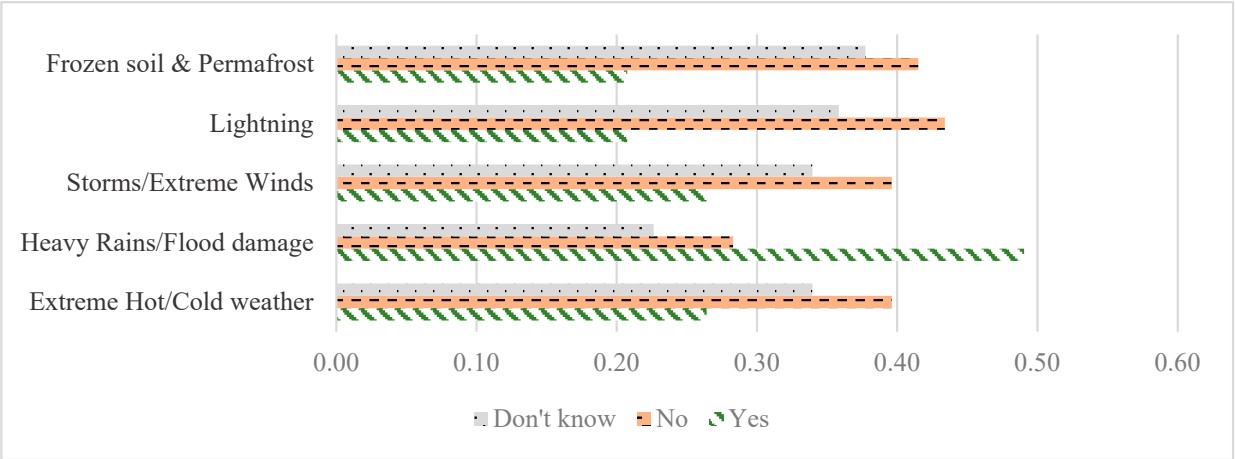

**Figure 20.** Performing specific assessment of weather and/or climate change factors in region/organization.

In connection with the previous question, respondents were asked about the approaches being implemented in their organization for assessing climate change impacts. The responses were provided as text, which was thereafter processed and categorized into five groups, as shown in Figure 21. The scores from the figure show that all the approaches have similar usage rates except for the GIS tool, which is not equally implemented to assess vulnerable areas in a transport network (Q24).

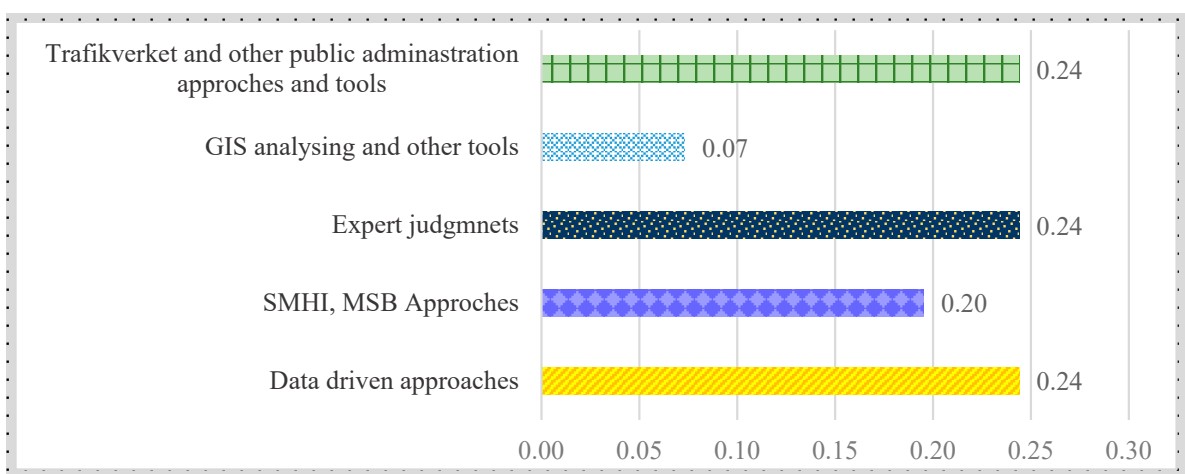

**Figure 21.** Implemented approaches for assessment of climate change impacts.

A detailed question was asked regarding whether a mechanism exists to assess current levels of risk for the identified vulnerabilities. Only 15% of respondents confirmed the existence of such a mechanism when performing their risk assessment. Figure 22 presents the comparative results between Sweden and UN. Like other questions, "Don't know/Blank" received a higher rank in both studies, which can be reduced by awareness acceleration within society. It is noteworthy that 60% of the respondents utilized risk and vulnerability analyses as mechanisms used for vulnerability assessment. There is also a significant difference between the vulnerability assessments of the two studies.

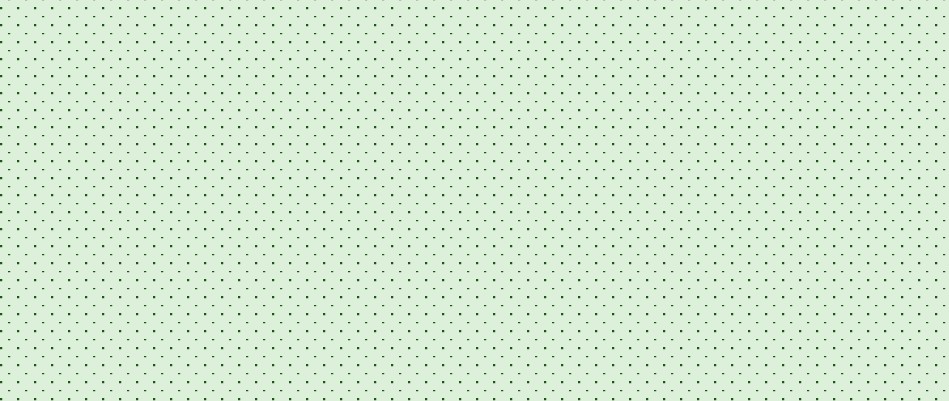

**Figure 22.** Following a mechanism to assess current levels of risk for the identified vulnerabilities.

### 4.5. Measures, Activities, and Policies for Climate Adaptation with References to Maintenance

The questions in this section evaluate the maturity level of climate adaption action among different stakeholders. In this regard, five distinct levels of adaptation measures have been defined (see y-axes in Figure 23). In connection to planning, 25 percent of the experts confirmed that their region has planned to undertake a general adaptation strategy to climate change impacts. Furthermore, 25 percent replied that they had "Already planned" and "Adopted and implemented" the strategy, and 50 percent replied, "Not applicable/Don't" know or "Not at all", as depicted in Figure 23 (Q29). Furthermore, this figure compares the penetration level of climate adaptation activity with a UN study.

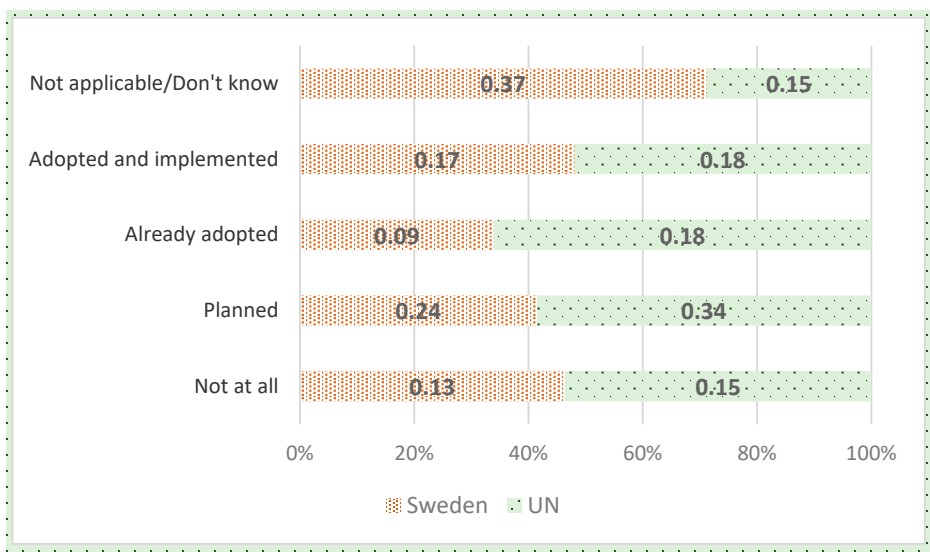

**Figure 23.** Ratio of adopted or planning to adopt a general adaptation strategy to climate change impacts (in general context).

Based on further investigation, different barriers to adaptation strategies (not specific to railway network) have been identified. Lack of resources, including skill, expertise, and time received a higher ranking, followed by political and prioritization issues and financial issues. Figure 24 represents the details of the barriers reported by experts (Q30).

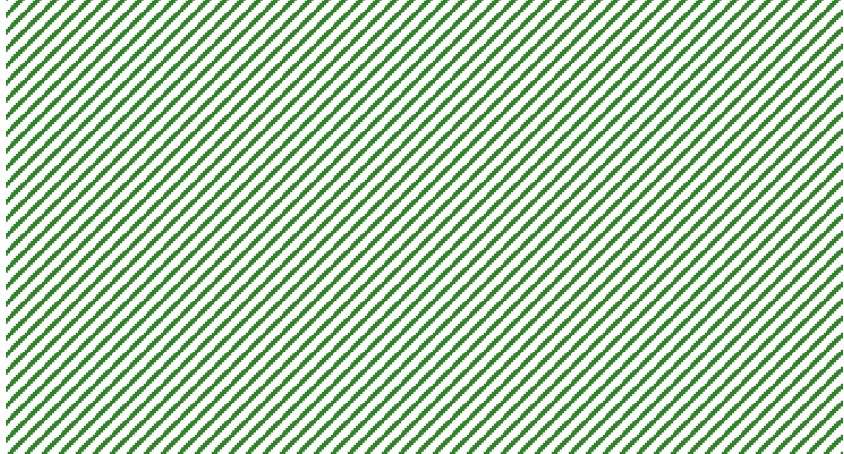

**Figure 24.** Reported barriers for general adaptation strategy.

In addition, the categorization of the above barriers in comparison with the UN study is illustrated in Figure 25.

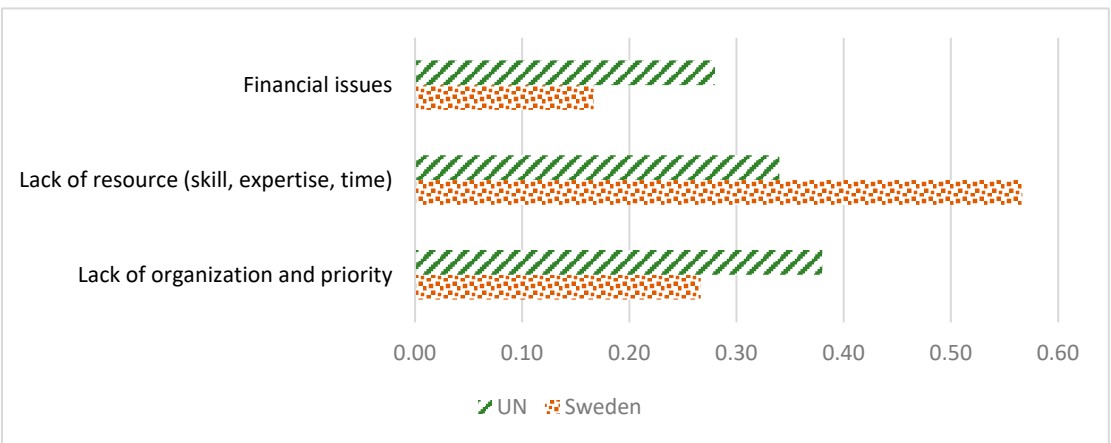

**Figure 25.** Comparison of the barrier between the two studies.

The Swedish Transport Administration manages the railway infrastructure and is responsible for climate adaptation and mitigation actions. This survey also revealed that the Swedish Transport Administration and Trafikstyrelsen were perceived as responsible public organizations dealing with climate adaptation and climate change issues for railway infrastructure. To identify the climate adaption strategy and its progress (i.e., adopted or planning to adopt) on railway infrastructure, there is a need to distinguish the respondents who are collaborating with TRV for railway infrastructure in their region. Hence, we have created two scenarios: (i) entire population and (ii) customized population, including those in collaboration with TRV. In the first scenario, 50 percent of respondents selected "Not applicable/Don't know" and only 20 percent reported a planned/adopted railway infrastructure-specific strategy for adapting to climate impacts (see Figure 26). In the second scenario, about 35 percent of respondents reported climate adaptation actions of railway infrastructure at various levels. In this scenario, 63 percent of respondents reported no climate adaptation actions for railway infrastructure in their region (see Figure 27).

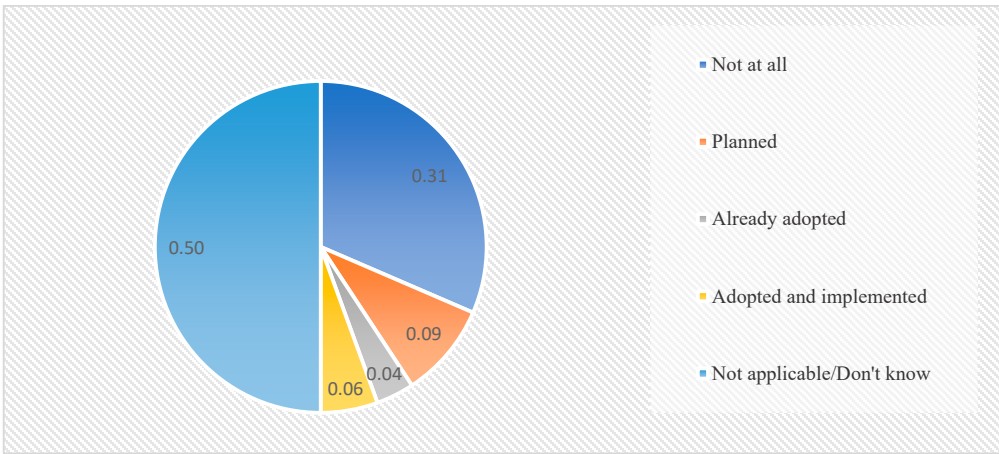

**Figure 26.** Adopted or is planning to adopt a railway infrastructure-specific strategy for adapting to climate impacts (scenario (i)).

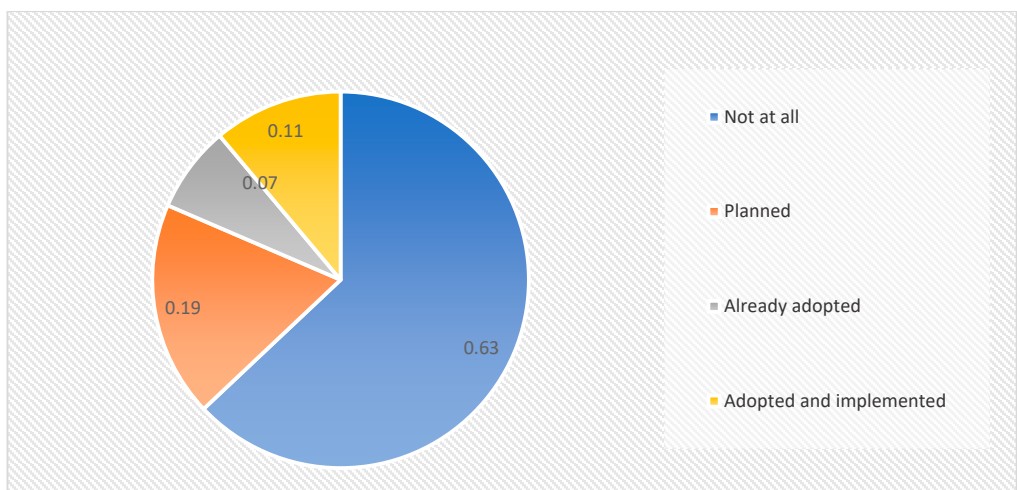

**Figure 27.** Adopted or is planning to adopt a railway infrastructure specific strategy for adapting to climate impacts (scenario (ii)).

When questioned about the climate adaptation barriers of railway infrastructure, only 40% of respondents wrote their experiences and opinions; such barriers are categorized into six different alternatives, as given in Figure 28. Organizational issues and time and resource restriction received a higher rate among other alternatives (Q32).

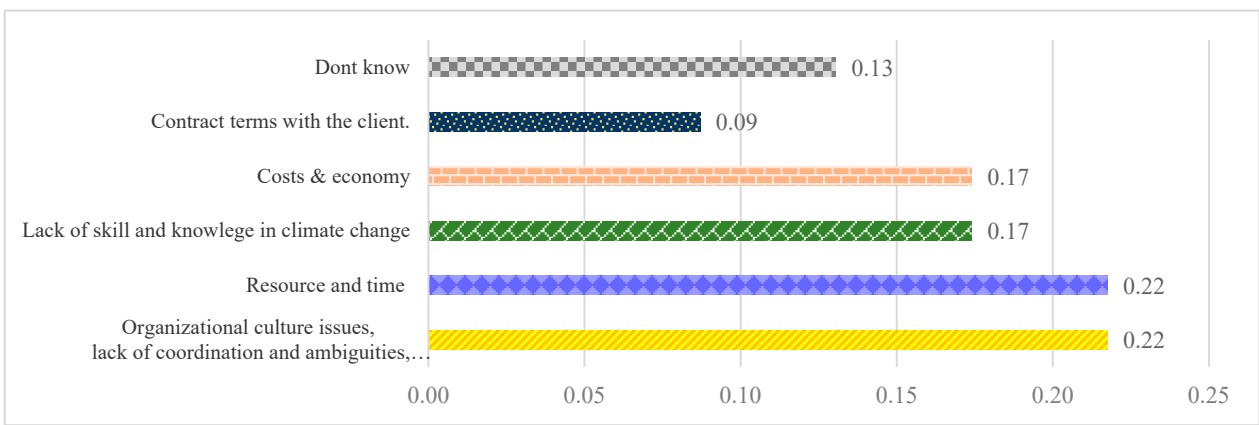

**Figure 28.** Reported barriers to adaptation of a specific strategy.

In the next steps, a question was asked about required actions that can be taken to improve the resilience of railway infrastructure to climate change impacts in their region. The outcome is summarized in Figure 29.

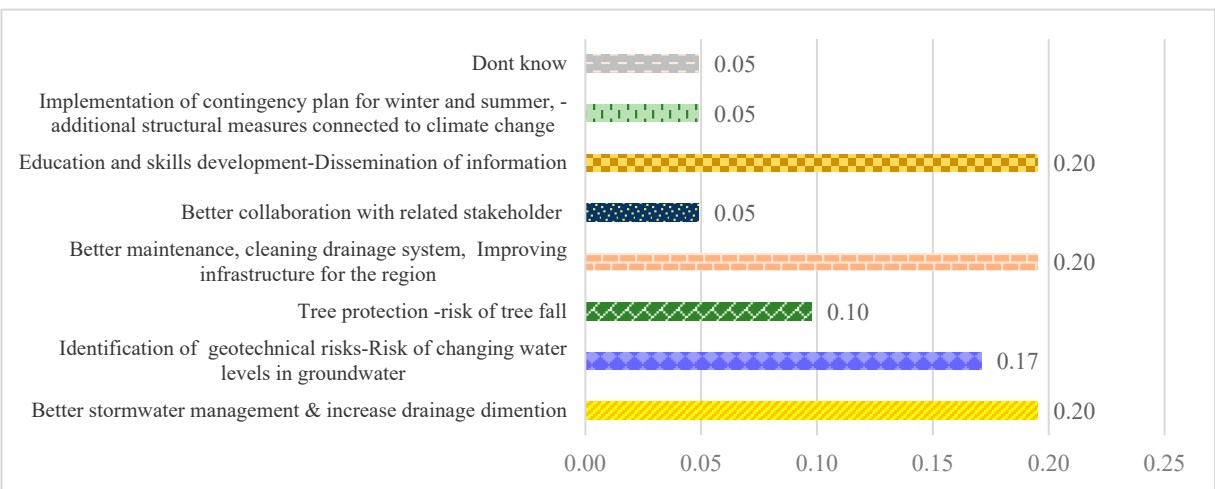

**Figure 29.** Required actions to improve the resilience of railway infrastructure to climate change impacts.

Among different alternatives, better stormwater management, better maintenance of drainage system, and education and skill developments had a higher rank in the analyses (Q33).

Related to the responses of the previous question, the inter-relationship of the above adaptation actions (given in Figure 29), including planning, investment, design, construction, operation, management, and maintenance, have been assessed. Planning, construction, operation, management, and maintenance were rated "High" and "Very high" by at least half of the respondents. The rate given by respondents for different railway infrastructure processes is illustrated in Figure 30.

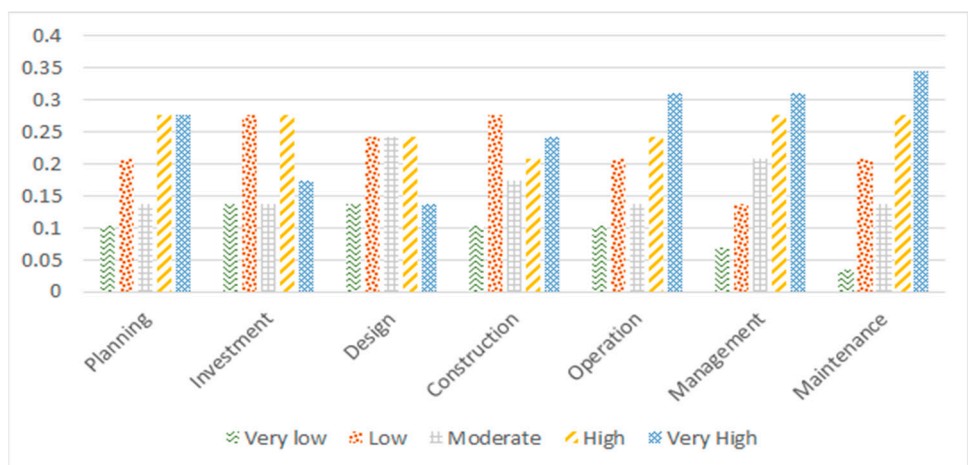

**Figure 30.** Inter-relationship ratio of the above-mentioned action to the railway infrastructure, planning, investment, design, construction, operation, management, and maintenance.

Furthermore, to compare with the UN study, the scaling was normalized by reducing the scaling category from five to two categories, low and high impacts. The outcome of the comparison is given in Table 3. To understand whether the adaptation policies are triggered when railway infrastructure is being undertaken/planned or not, a question was asked to identify which climatic impacts have been considered. The outcome is depicted in Figure 31.

**Table 3.** Comparison of studies for action to improve the railway infrastructure, planning, investment, design, construction, operation, management, and maintenance.

|  | Impact Level | Planning | Investment | Design | Construction | Operation | Management | Maintenance |
|---|---|---|---|---|---|---|---|---|
| UN case | Low Impact | 0.35 | 0.5 | 0.4 | 0.55 | 0.6 | 0.5 | 0.55 |
|  | High Impact | 0.65 | 0.5 | 0.6 | 0.45 | 0.4 | 0.5 | 0.45 |
| Sweden case | Low Impact | 0.45 | 0.55 | 0.62 | 0.55 | 0.45 | 0.41 | 0.38 |
|  | High Impact | 0.55 | 0.45 | 0.38 | 0.45 | 0.55 | 0.59 | 0.62 |

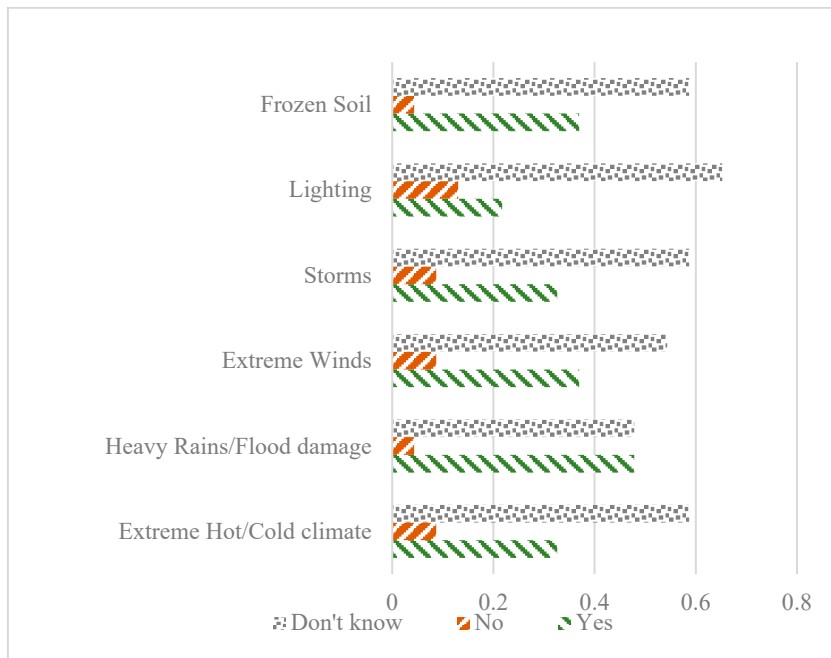

**Figure 31.** Consideration of climate change impacts for developing new railway infrastructure.

For this question, about 50 percent of respondents answered "Don't know" from municipalities and other stakeholders who did not receive information on railway adaptation policies. Except for this group of experts, most of the respondents believed that different adaptation policies had been undertaken when developing a new railway line, and only a few believed that there were no adoption policies for such developments (Q35).

Climate change and its associated impacts will continue for many decades, and even centuries, regardless of the success of global initiatives to reduce greenhouse gas emissions [1]. Hence, there is a need for emergency response systems dealing with vulnerabilities affected by changes in climate conditions. Concerning the development and planning of emergency response systems for the railway sector, most of the respondents (about 50%) selected the "Don't know" alternative and about 20 percent said they had "Planned" or "Already adopted, implemented" emergency response systems for the railway infrastructure and operation for disasters due to climate change. Figure 32 represents the assigned rate by experts for different alternatives (Q36).

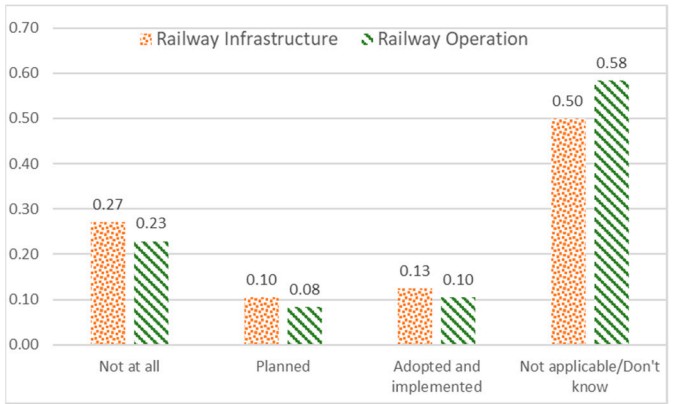

**Figure 32.** Plan for developing emergency response systems of railway due to climate change disasters.

Furthermore, respondents pointed out several alternatives to identify significant areas that need further attention to enable effective adaptation strategies in railway infrastructure tailored to local conditions. Maintenance of infrastructure was prioritized by 20 percent of respondents, followed by required adaptation activity for flooding and drainage systems. The other reported alternatives are plotted in Figure 33. Based on the identified significance areas, an important inference is a need for further research and study on the risks of climate change and their impacts on the maintenance of infrastructure and adaptation measures (Q37).

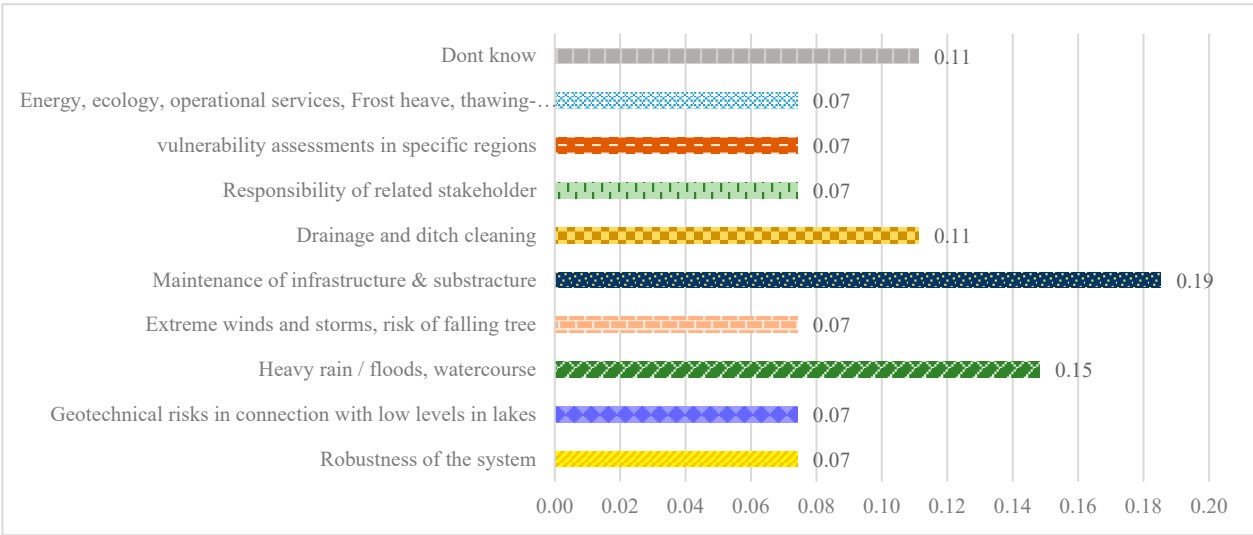

**Figure 33.** Priority areas that require further attention to enable effective adaptation strategies in railways.

Despite the negative impacts and dimensions of climate change on our society, there is an opportunity that may arise from climate change. Hence, it is interesting to identify the potential opportunities for the railway sector based on the respondents. The tourism industry is expected to increase, followed by growth in mining and agricultural production. Moreover, a reduction in snow removal tasks and their related costs are considered as opportunities for railways that may arise in response to climate change. Some respondents identified other opportunities, such as the new demand for rail transport and the reduction of driving and transport by truck. The rate of other potential impacts is illustrated in Figure 34 (Q38).

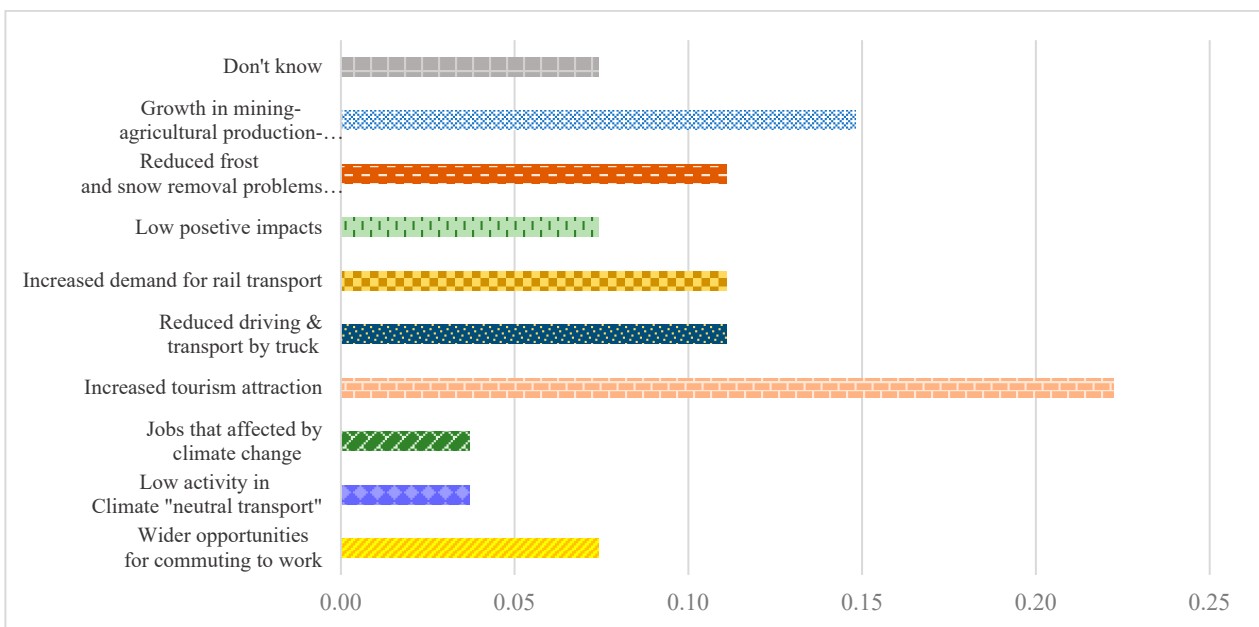

**Figure 34.** Opportunities that may arise for the railway sector in your region in connection with climate change.

*4.6. Evaluation of Adaptation and Maintenance Debt of Transport Infrastructure*

The last stage in Section 4.5 is to map the current level of actions and activities, including technical, political, administrative, etc., with references to adaptation and maintenance in the transport network.

The first step is to determine the time horizon to achieve robust infrastructure in advance, considering the financial and resource availability, criticality of assets, etc.

The second step is to identify the status of adaptation and maintenance activities in the area under study for the specific asset. The outcome of the questionnaire is to support the transport administration experts to identify the vulnerable areas and related adaptation barriers. In addition, it provides a better understanding of the status of action and its level of adaptation, whether the activity is "Already planned" or "Adopted and implemented", or even if the respondent answered "Not applicable/Don't know". This approach helps infrastructure managers keep track of progress and update the status of adaptation and required maintenance to reduce the adaptation and maintenance development debt.

## 5. Conclusions and Remarks

Climate change will adversely affect the operation, safety, and maintenance of railway infrastructure if no remedial action in terms of climate mitigation as well as climate adaptation is implemented. The projected long-term rise in the mean air temperature will lead to an increase in frequency and intensity of extreme weather events which will, in turn, affect the susceptibility of railway infrastructure. For example, extreme rainfall will lead to damage of infrastructure and flooding of urban areas if remedial actions such as enhanced drainage systems with a higher capacity to drain water are not adopted. Using a questionnaire approach, the study assessed how the level of awareness of climate change impacts on the Swedish railway infrastructure is perceived. It also mapped the causal relationship between climate parameters and infrastructure conditions and identified the risk incidences connected to climate change.

The study reveals that the level of awareness of climate adaptation options needs to be raised. Thus, there is a need to define and formulate regional policies for climate change adaptation of railway and road networks, including awareness-raising and sharing best practices. It is recommended to share the outcome of studies from different countries with the various departments within governments and other stakeholders to plan for adaptation options.

Based on the questionnaire responses, we found that most of the respondents are aware of the impacts of temperature, heavy floods, winds, lightning, and permafrost on railway operation, safety, economic impacts, and robustness. However, it was agreed that additional work is required to quantify those effects and impacts on various railway assets for each critical location so that appropriate adaptation measures can be planned and implemented.

We found out that most respondents from Sweden are aware of the availability of climate data sources and models not only from SMHI but also other sources where they can glean more insights and understanding of climate change in both short-term and long-term periods. However, most of the respondents are not utilizing the full potential of data because of unawareness of existing tools, cost of tools, and knowledge of working on data.

Regarding railway infrastructure risk incidences associated with climate change, only a few assessments have been carried out to identify types of risks and their associated consequences. The predominant risk incidences identified in the study are rail buckling, bridge scouring, signaling system failure, and inadequate drainage capacity.

The maturity level of climate adaptation action differs within the organizations involved in this study. Fifty percent of the respondents confirmed that their organizations have planned or implemented a general adaptation strategy to climate change impacts. Among the strategic adaptation solutions mentioned by the respondents are better stormwater management, better maintenance of drainage system, education, and skill developments. Lack of knowledge and resources, lack of coordination between organizations, and large investment costs are some of the barriers to the implementation of climate adaptation strategies.

The results of this study show that effective climate adaptation approaches are required to reduce climate and maintenance debt, including awareness, risk mapping, vulnerability assessment, maintenance, and emergency planning. Finally, introducing effective guidelines and regulations for railway infrastructure design and construction considering climate change parameters and utilizing new condition monitoring technologies and systems are essential actions required for climate adaptation and emergency response systems.

**Supplementary Materials:** The questionnaire is available online at https://forms.office.com/Pages/ResponsePage.aspx?id=i0BTVM2mHkyLEBi1APtUTm4q2DwnvUxIksZQdrg7DHxUMk03WFI3TjU5OVY2OUhZTkJRNlVPUlNQNC4u (accessed on 18 November 2021).

**Author Contributions:** Conceptualization, A.H.S.G. and A.T.; methodology, A.H.S.G.; software, A.T.; validation, S.F. and U.K.; formal analysis, A.H.S.G.; investigation, S.F., A.H.S.G. and A.T.; resources, A.H.S.G., A.T. and S.F.; data curation, A.H.S.G. and A.T.; writing—original draft preparation A.H.S.G.; writing—review and editing, A.T., S.F. and U.K.; visualization, A.H.S.G.; project administration, A.H.S.G. All authors have read and agreed to the published version of the manuscript.

**Funding:** The authors gratefully acknowledge the funding provided by Sweden's innovation agency, Vinnova, to the project titled "Robust infrastructure–Adapting railway maintenance to climate change (CliMaint)". The funding was granted in a competitive application process that assessed replies to an open call for proposals concerning "Innovations for a sustainable society (IHS)—for a climate-neutral future" (grant no. 2019-03181).

**Institutional Review Board Statement:** Not applicable.

**Informed Consent Statement:** Not applicable.

**Acknowledgments:** The authors gratefully acknowledge the technical support and collaboration (in-kind support) of the Markus Lundkvist from Trafikverket (Swedish Transport Administration), Lina Andersson from Trafikstyrelsen, and Veronica Jägare from Luleå Railway Research Center (JVTC), Gustav Sandberg from SMHI, Per Norrbin from Sweco Rail AB, Roland Bång from InfraNord, Patrik Ruumensaari from Luleå Municipality, and all other municipalities in Sweden.

**Conflicts of Interest:** The authors declare no conflict of interest. The funders had no role in the design of the study; in the collection, analyses, or interpretation of data; in the writing of the manuscript, or in the decision to publish the results.

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
