# Peer review of "Adapting Railway Maintenance to Climate Change"

_sustainability, doi:10.3390/su132413856_

Round 1
Reviewer 1 Report
In this paper, the authors used a questionnaire as a tool to qualitatively identify and assess the impact of 17 climate change on railway infrastructure with associated risks and consequences, and mapped the causal relationship between climate parameters and infrastructure conditions and identified the risk incidences connected to climate change.
Major:
1.How is the parameter setting in the questionnaire survey method determined, and whether the respondents are representative.
2.Part of the pictures in the article are in the form of radar chart, for example, Fig. 10, while other parts are in other forms. Is it for aesthetics or more straightforward explanation?
3.In this paper, the selected parameters are fully analyzed, but how to deal with the relationship and relevance between the parameters?
Minor ones:
- Line 169, P4, 'relationship' should be 'relationships'
- Line 306, P9, 'have' should be 'has'
Author Response
Dear Editor/Reviewer
We would like to thank the reviewer for their support and constructive comments which increased the readability of the paper.
We have answered all the questions that we have received from the Reviewer,
Please find the attached file.
Regards
Amir Garmabaki

Reviewer 2 Report
In this paper, the authors explore the impact of various climate changes on railway infrastructure with associated risks and consequences. A qualitative approach based upon questionnaire is utilized to analyze the level of awareness about the impact of climate changes. Several beneficial suggestions and policy recommendations are proposed to improve the resilience towards climate change. Overall, the paper is well written and easy to understand. The reviewer has only a few questions or concerns:
The authors could improve their paper by more clearly elaborating on its contributions in relation to the existing literature. It would be better to introduce climate adaptation in other regions such US or Asian countries.
Section 2 describes adaptation and maintenance development debt of transport infrastructure. In my opinion, the authors should explicitly describe its relation with Section 3. In other words, how to use the results of adaptation and maintenance development debt in Section 2?
One relevant reference is missing, that is Gu, Y., Fu, X., Liu, Z., Xu, X., & Chen, A. (2020). Performance of transportation network under perturbations: Reliability, vulnerability, and resilience. Transportation Research Part E: Logistics and Transportation Review, 133, 101809. It employs a quantitative method to evaluate transportation performance under perturbations, such as extreme weather events in this study.
Author Response

(The authors gave the same response as above.)

Reviewer 3 Report
This study aims to examine the impact of climate change and extreme climate on the operation and maintenance activities of the Swedish railway infrastructure. The purpose of this study is very interesting. Some issues that the authors should address in order to improve their work are as follows:
- The authors state that this work used qualitative analysis. I think there is quantitative analysis such as descriptive statistics, radar charts, and reliability analysis.
- The 38 questions should be reported in Appendix. I think the provided URL will be disappeared.
- Provide more details of the questionnaire, such as a number of parts and a brief of each part.
- The authors report that 55 responses were received. How many target respondents of the research design that enough? Give evidence of enough respondents.
- Line 203-204, “The term “expert” used in this study is according to 203 the definition given in [63, 64]”. Please check the intext citation format; I think it should be Authors [XX].
- Please check the total number of a participant in Table 1 that is not 55.
- The methods of data collection were not clear. Does this work use all these methods, in-depth interviews, focus groups, questionnaires, or not? Please provide more details.
- The reliability of the questions with Cronbach’s alpha should be reported in all constructs. Moreover, provide the cut of the value of this value.
- Most of the qualitative analysis reported the opened-end question and/or experts’ opinion, but I am not found.
Author Response

(The authors gave the same response as above.)
